# "It's my life, not theirs!" Therapeutic itineraries and refugee reflections on referral health care in western Tanzania

Zachary Obinna Enumah [1,2]*

1 Center for Global Surgery, Department of Surgery, Johns Hopkins Hospital, Baltimore, MD, United States of America, 2 Department of International Health, Johns Hopkins School of Public Health, Baltimore, MD, United States of America

* zoe@jhmi.edu

## Abstract

Globally, refugees number over 25 million. Yet, little attention has been paid to how refugees access referral health care in host countries. By referral, I mean the process by which a patient deemed too sick to be managed at a lower-level health facility is transferred to a higher-level facility with more resources to provide care. In this article, I provide reflections on referral health care from the perspective of refugees living in exile in Tanzania. Through qualitative methods of interviews, participant observation, and clinical record review, I trace how global refugee policy on referral health care manifests itself in the lives of refugees locally in a country like Tanzania that has strict policies and limitations on freedom of movement. In this space, refugees experience complex medical problems, many of which began prior to or during their flight to Tanzania. Many refugees indeed are approved to be referred to a Tanzanian hospital for further treatment. Others are denied care or pursue other therapeutic itineraries outside the formal system. But, all are subject to policies of Tanzania that restrict freedom of movement and almost all experience delays on several levels (e.g., waiting for a referral, waiting at the referral hospital, waiting for follow-up appointments). In the end, refugees in this context emerge not simply as passive beings upon which biopower is enacted, but also as active agents, sometimes circumventing a system of power in their pursuit of their right to health, all in the context of strict policy that seeks to enforce state security over one's right to health. In the process, refugee experiences with referral health care become a window into the larger politics of refugee hosting in Tanzania in the present day.

## Introduction

"Why are you doing this to me?"

These are the words that Thomas—husband, pastor, father of eight, and refugee from the Democratic Republic of Congo—asked the head administrator of the main health center in Nyarugusu refugee camp in western Tanzania.

Thomas is a refugee who fled with his family from violence in North Kivu, DRC. He was shot in the leg while in the DRC, and since then has had an ongoing issue with a chronic,

**Data Availability Statement:** Data used and analyzed in the current study are not publicly available due to privacy and personally identifiable health information that could compromise the

participants who were interviewed in the study. Data could potentially be de-identified upon reasonable request but data was obtained in Kiswahili language. Data are available upon request from the IRB via email (jhmeirb@jhmi.edu) with attention to Committee Chair, Susan Bassett, for researchers who meet the criteria for access to confidential data.

**Funding:** This project was supported by Association for Academic Surgery Global Surgery Research Fellowship (ZOE) and the American Society of Tropical Medicine and Hygiene Ben Kean Fellowship (ZOE). The funders had no formal role in study design, data collection and analysis, decision to publish, or preparation of the manuscript.

**Competing interests:** The authors have declared that no competing interests exist.

draining, non-healing wound and pain. This has been ongoing for about ten years. "I don't have peace," he says, as he continued to have pain and lose sleep at night because of his leg.

After arriving to the refugee camp in Tanzania, Thomas went to one of the health posts run by a humanitarian organization that has been working in the camp for several years. After several trips to the health centers in the camp, he found one doctor who told him he would be referred to one of the zonal hospitals in Tanzania, Bugando Medical Centre (BMC). Normally, for a patient to receive and physically be referred to another hospital for care, the patient is first seen by a doctor who decides in their clinical judgment that the patient cannot be treated with the resources at the refugee camp and thus needs to be referred to a higher level of care. The patient's case is then presented before a Medical Referral Committee comprised of United Nations High Commissioner for Refugee (UNHCR) staff, NGO staff, and physicians. For transportation efficiency, referrals are often grouped together and several refugee patients may travel at once to the same hospital. On the day of his referral, Thomas packed his clothes and went to the main hospital to be transported to the zonal hospital. Upon arriving there with his fellow patients, he was told the referral was no longer happening and that he should return home and wait to be called again. "We didn't have an option apart from going back home and waiting," he said. But, he was told he would be informed of the next date of departure through one of the community health workers in the camp. Thomas waited.

"I continued going to the main hospital often to tell them that I have pain and that I am not sleeping, and also that I, too, am a human being and I need to be treated for my condition." Thomas was told they would send a community health worker, but no one visited Thomas to inform him of the departure date for his referral. On the new scheduled day of departure at the main hospital, they read Thomas' name from a list of patients, but he was not there because no one had informed him, and no one had his cell phone to call him and tell him to come. The others went. Thomas was left behind, and Thomas continued waiting.

Thomas was told he would have to wait now until another Medical Referral Committee (MRC) meeting occurred where decisions are made about other referrals and that he would be considered on that trip. He was told the meeting would happen in June or July. We spoke on the last day of the month of July and there was still no word of his referral. Thomas continued to wait.

## Background, setting and context

### Refugee health care

Globally, there are over 26 million refugees in existence today [1]. Forced migrants including refugees face significant challenges in accessing health care, including language barriers, lack of knowledge of information about local health systems or disease outbreaks, lack of financial resources or stability, and discrimination [2–4]. Research suggests that refugees face the triple burden of health conditions including mental illness, infectious diseases, and non-communicable diseases [5].

Countries take various approaches to hosting refugee populations across the globe. The United Nations High Commissioner for Refugees (UNHCR), the main UN body in charge of refugee issues, outlines three durable solutions for refugees: 1) repatriation; 2) local integration; or 3) resettlement to a third country [6]. Some countries, such as Uganda and Lebanon, allow for more freedom of movement. Other countries, such as Tanzania and Malawi, have more strict refugee policies that encourage or mandate forced encampment of refugees [7]. The distinction is important because the macropolitical choice influences the micropolitics of health care and care-seeking. In countries that do not require refugees to live in specific areas or camps, there is generally greater freedom of movement, and this can aid in access to and incorporation of refugees into national health systems, albeit there is an increased financial

pressure on payment for services [8]. In the forced encampment model, such as in Tanzania, refugees are often provided more access to health services at some level and often free of charge. But, access to higher levels of care is problematized given refugees are not allowed to move freely outside of the refugee camp without special permission.

Tanzania has a complex history of hosting refugees over the past six decades. In general, the country transitioned from a socialist government in the 1960s to a neoliberal one in the 1990s, and in doing so transitioned from an "insurer of equity and distributor of resources to a facilitator of trade liberalization, privatization and [foreign direct investment]" [9]. A focus on privatization and foreign investment in the wake of structural adjustment programs that were pushed on many reluctant African governments contributed to a stark change in its refugee policy. Freedom of movement has become essentially obsolete, and a permit is required for any movement out of the refugee camp. In Tanzania's current National Refugee Policy (2003), local integration is not mentioned as a durable solution. Recent events in Tanzania suggest the country is even less invested now in hosting refugees and more focused on state security. In addition to the well-documented forced repatriation of Rwandan refugees in the 1990s, other recent examples include the withdrawal of Tanzania from being a pilot site for the Comprehensive Refugee Resettlement Framework and the forced return of Mozambican refugees fleeing violence in North Mozambique [10–12].

## Ecology of referral health care

Referral health care requires the movement of individuals from one level of the health care system to another. In general, referral health care has been defined as a hierarchical system that "permits movement of patients from the base of the national health care system to its apex and vice-versa" [13]. In other words, national health care systems are often hierarchical or vertical in nature, and patients can be transferred from one health care center to another one that may be more specialized or better equipped to manage a particular illness. More specialized health care (e.g. specialty surgery, cancer treatment) is often available at fewer, higher-level centers that are often located in urban areas. Yet, in general even rural populations have theoretical access to such levels of care through a process of referral. If a patient presents to a primary health care center and is unable to be treated, they may be referred to a higher level center, such as a district, regional or national hospital.

Tanzania has three levels of health care: primary, secondary and tertiary. The primary level is comprised of health dispensaries and health centers, which offer basic medical and surgical services, such as vaginal or uncomplicated caesarean sections, minor burn and wound care, and malaria testing, to name a few examples [14]. Secondary health centers, such as district level hospitals, may offer additional services, such as more general surgical procedures (e.g., hernia repair, appendectomy). In Tanzania, tertiary centers are both zonal level hospitals and the national hospital, Muhimbili National Hospital, where services such as cancer treatment or heart surgery may be offered. The premise is that even a patient who presented to a lower-level center may be referred up the chain one or more times so as not to overburden the zonal and national hospitals, since some patients may also self-present. A strong referral system is a cornerstone of any high functioning health care system.

In this article, I provide narratives of referral health care from the perspective of refugees living in exile in Tanzania. In taking one approach of critical medical anthropology in examining structures of power and how inequalities and inequities of health can often best be understood by listening to the stories of the very individuals who live and suffer from them, this article seeks to add to the literature at the intersection of migration and health, with a particular focus on referral health care [15].

Referral health care generally refers to the movement of patients between different levels of a health care system to optimize care for patients, most commonly with more subspecialized or advanced services offered at fewer, higher-level centers. The concept and research on referral health care often takes for granted the right to freedom of movement, yet refugees in Tanzania do not freely enjoy freedom of movement. Based on several months of fieldwork between 2011 and 2021 and qualitative methods of interviews, participant observation, and clinical record review, I trace how global refugee policy on referral health care are translated and experienced locally in a country, such as Tanzania, that enforces strict policies and limitations on freedom of movement.

Previous literature has explored refugee population utilization of host country health services including for non-communicable, chronic, or surgical conditions such as cancer, where barriers such as cost are often cited [16–20]. Other scholarship has focused on participatory assessments—that is beneficiary perspectives—of general health services [21].

However, little attention has been paid to the refugee perspectives, attitudes, and opinions of health processes such as referral. Unique questions arise at the nexus of refugee health and politics. How do refugees navigate a complex space to seek health care when their freedom of movement is restricted? What do they do when the health care system while offered free to them fails to meet their health needs or the needs of their families? What systems of power influence, regulate, discipline and control refugee movement and what effect does that have on their health?

In this space, refugees experience complex medical problems. A deeper examination of the referral process from the refugee perspective highlights rate limiting steps in the process of referral. In this space, health conditions—particularly surgical ones—become a window into larger politics of movement and freedoms for refugees in Tanzania. Many refugees indeed are approved to be referred to a Tanzanian hospital for further treatment. Others are denied care or pursue other therapeutic itineraries (see *Theoretical Framework* below) outside the formal system. But, all are subject to policies of Tanzania that restrict freedom of movement. Almost all experience several levels and layers of delays in waiting for a referral. Such restrictive legal policies create a unique problem-space whereby refugees seeking health care face an additional threat not only of physical illness but also of political fear—on top of the fear that caused them to leave their country of origin. In the end, refugees in this context emerge not simply as passive beings but also as active agents in pursuing their own right to health in the context of strict, policy that seeks to enforce state security over one's right to health.

## Study site and background on referral for refugees

Nyarugusu refugee camp was established is 1996, and it is located in Kigoma region of Tanzania. It is home is approximately 130,000 refugees from the Democratic Republic of Congo and Burundi. It is the largest of three refugee camps in Tanzania. The United Nations High Commissioner for Refugees (UNHCR) and the Tanzania Ministry of Home Affairs (MHA) regulate provision of services and administrative functions of the refugee camp. The UNHCR partners with several national and international NGOs to provide basic services, such as food, water, sanitation, and health services. The MHA governs camp permits for entry and exit and regulates who and what can enter the refugee camp. Regarding health, the medical services are run by humanitarian organizations in the camp. The camp supports one main dispensary health center (akin to a hospital), two other health posts where patients can be admitted overnight, and multiple other health posts that serve refugees and local Tanzanians who live in the surrounding area. The main health center houses two major operating theatres and one minor operating theatre. Approximately 200,000 people are served at the health centers because the

center also serves Tanzanians from surrounding villages. The camp-based services are free for both refugees and Tanzanians.

## Referral health care in the refugee camp

Since the refugee camp health center is a primary level facility, it is common for patients to present to the hospital for medical or surgical reasons and the hospital does not have the resources to diagnose or treat. For those patients deemed untreatable at the camp level hospital, a referral is made possible. Typically, this involves a treating physician seeing a patient and deeming they require a referral to a higher level of care for diagnostic or therapeutic purposes. A physician would formally write a referral on a standardized form detailing demographic characteristics, the patient's history of present illness, hospital course including medications/treatments, investigations performed, and reason for referral. Then, the case would be presented for discussion at a monthly Medical Referral Committee (MRC) meeting which is attended by representatives of various NGOs, the UNHCR, and the government. On paper, the decision to refer a patient is based on local and international guidelines with decisions often made focused on prognosis and cost [22, 23].

In the space of referral, however, complex questions remain on how a refugee population can access such higher levels of care if their medical problems cannot be treated at the refugee health center level and if Tanzania's refugee policy does not permit freedom of movement, as refugees needs special permits to leave the refugee camp. How do refugees navigate this complex problem-space? What therapeutic itineraries (see *Theoretical Framework*) do they embark on and what guides their decisions? In this article, I seek to answer these questions by providing an often overlooked experience—the refugee perspective—by approaching such questions through existing anthropological theories.

## Biopower and therapeutic itineraries

Anthropology has long been focused on better understanding relations of power and politics on a macro and micro level. The power of anthropology and ethnography in global health often focuses on the social and political aspects of health care, including the interaction between the two [24, 25]. In this article, I draw on *biopower* and *therapeutic itineraries* as anthropological theories through which to understand the intersections of referral health care and refugeehood. Originally coined by Michel Foucault, biopower refers to how on a population level, "biological and medical data are used by institutions. . .[to] *discipline* populations" [26]. In doing so, institutions (and the individuals who run them) possess literally a "power over life" and may decide who lives and who dies [27]. One specific branch of biopower is the concept of *therapeutic citizenship*, whereby an individual may make claims on their right to health based on having a certain illness. Two recent examples of this include Nguyen's work on HIV/AIDS in west Africa and Petryna's work on radiation victims in the wake of Chernobyl, where individuals make claims to the State for benefits as a result of having radiation sickness [28, 29]. In other words, certain diagnoses may grant someone access to certain health resources.

Appreciating the distinction anthropologists have made between *disease* (a biomedical or practitioner's understanding of pathology) and *illness* (a patient's holistic experience of their medical problem), social scientists including anthropologists have also long been concerned with the patient experience of their illness. The concept of *therapeutic itineraries* refers to a patient's trajectories (real, imagined, or aspirational) they pursue to seek medical treatment, as well as the meaning placed on their illness and suffering, amidst particular sociocultural, economic, and political circumstances; these therapeutic routes, journeys, or itineraries may not

use pre-determined biomedical terms [30, 31]. Therapeutic itineraries are the "route[s] taken by individuals or groups, in order to preserve or restore health" [32]. Given the appreciation for sociopolitical circumstances within the larger concept of therapeutic itineraries, it is a useful framework from which to understand perceptions and experiences of referral health care among refugees.

## Methods

### Sampling, data collection and analysis

In this article, I draw on several months of field work conducted over approximately six non-contiguous months between 2011 and 2021. The primary ethnographic methods used for this paper included interviews, participant observation and record review, and a triangulation of the aforementioned data methods. Interviews were conducted with refugees primarily between July and September 2021. Participants for interviewing were selected by reviewing over 200 clinical referral records. On average, the Medical Referral Committee (MRC) discuss about 100–150 patients during each meeting. The referral records were prepared for any patient whose case was to be presented in front of the MRC, where decisions are made on which patients are approved for referral. Each clinical record contained information on patient demographics (age, sex, religion), presenting symptoms, hospital course, investigations and laboratory tests performed, reason for referral, and expected prognosis. I carefully reviewed over 200 of these records independently and attended a medical referral committee meeting. Specific cases requiring referral were thus selected to interview, although not all of those selected were available or reachable. In terms of participant observation, my time in the camp was both as a practicing clinician (seeing and referring patients myself in clinic) and researcher. This both influenced the spaces I had access to (e.g. morning rounds/report, operating theatre, outpatient clinic, inpatient wards) and my interpretation of ongoing phenomena (e.g. whether a patient needed a referral in my own clinical opinion). My time in Nyarugusu also included visiting research participants in their homes for interviews. A Tanzanian research assistant led the interviews using a semi-structured interview guide, and I was present for interviews in July and August 2021—accounting for about 25% of the total of 20 interviews conducted. Interviews focused on personal experience of the patient or family member, and interviews also contained what Morse (2000) has called shadowed data, where an individual can speak in detail about another's experience [33]. The research assistant is a clinical officer in the Tanzania health care system and had previously worked in the camp as a health care provider. Thus, he was intimately familiar with the referral process and common disease pathologies afflicting camp residents. He underwent training in research methodology and assisted in co-leading a separate ethics/research training workshop for other health care educators as part of a larger, survey study in the camp. Interviews were conducted in Kiswahili, the most common language spoken in the camp and the national language of Tanzania; most were transcribed into Kiswahili text, and then majority of those were translated into English text. I have fluency in Kiswahili and English. This article is based on a total of 18 interviews with refugees, as two of the total 20 interviews were with Tanzanian patients and excluded for the purpose of formal analysis.

Thematic analysis was performed on the data through both deductive and inductive coding methods. Coding was deductive by using the theoretical frameworks described above as guides for interpreting the findings, transcripts, and observations, while also being open to other themes that may be emerging from the data itself in line with grounded theory methodology [34]. Computer software, Nvivo, was used to assist with coding and data management. Field notes and/or memos were written throughout the research process. All names and locations used in this paper are pseudonyms or anonymized.

### Positionality and ethical clearance

I am a clinician at an academic medical center in the United States. I am also a licensed physician in Tanzania. During my fieldwork, I actively participated in caring for patients, including referring patients to higher levels of care, attending an MRC meeting, and even personally escorting and caring for one patient during the referral process who later died. This work is personal to me, and the challenges I faced as a physician in the camp may also come to light in my perceptions of what I observed around surgical and referral health care. My own clinical opinion undoubtedly influenced my observations about which cases I felt merited referral or which ones may not have. My status was as a volunteer clinician and researcher under the umbrella of a humanitarian organization, and I relied heavily on local expertise and guidance in understanding local context and decision-making. By my nature of being a health care provider in the camp, I was also privy to conversations or situations around the concept of referral, and these have both consciously and subconsciously influenced the ways in which I interpret the data I present in this article. I present this positionality to provide context from which I draw my own conclusions as expressed in this article.

Taken together, this paper draws on the approach in critical anthropology of combining research methods to better seek a co-created truth (co-created subject and researcher. In line with social scientific approaches to understanding validity and reliability, I suggest that the various methods used in this paper (interviews, participant observation, and document review) allow for a better understanding of a idea of how referral works—or does not work—from the perspective of refugees in this particular location [35]. While these phenomena may not be generalizable to a broader context of other refugee situations throughout sub-Saharan Africa or the world, I hope that the combination of multiple methods of inquiry and the triangulation of said methods allows for a rich picture of referral health care in this particular context.

Research approval was obtained from the Johns Hopkins Medicine Institutional Review Board (IRB00258009), as well as the Tanzanian Commission on Science and Technology (2020-391-NA-2011-143). The Ministry of Home Affairs of Tanzania granted a permit to enter the refugee camp. Permission for interviews (e.g. consent) for the research was obtained verbally from participants or an appropriate guardian/adult and this was documented in the transcript(s).

## Results

Findings from interviews, observation, and clinical record review reveal several key themes. First, medical and surgical problems of refugees are complex and many began before or during the flight process. Additionally, there are several levels and layers to delays in referral health care. Lastly, the health care seeking behavior of some refugees involve a mix of independent care seeking and circumventing the existing system, while others fear to do so and wait for significant amounts of time.

### Complex medical problems

Joseph sat peacefully in a wooden chair, TV behind him, wearing a neon blue shirt hiding his abdominal binder underneath. Unassuming, soft-spoken, but to the point, Joseph, a 23-year-old Congolese refugee who came to Nyarugusu as a young baby, has lived more than 20 years in Nyarugusu camp. He has HIV, heart failure, and a large abdominal hernia—a defect in his abdominal wall where his inner contents can protrude through. He recounted his story of accessing health care and referral services at the camp.

"In 2019, my condition started getting worse. Previously, I was ill, but not that serious. Suddenly, the condition changed, and I started having swelling on my feet and abdominal swelling. At the hospital when they performed tests, nothing was seen. One day a doctor. . .. finally said I was having liver problems so I have to be referred. But, I took a long time to go to the hospital. I was written to go, [but] it was last year in June (2020) when the referral was written but I went there this year (2021)."

Joseph was first referred to Kabanga hospital, a district hospital about 65km from the refugee camp. At the time, there were visiting doctors from the specialist zonal hospital (a process sometimes referred to as "reverse referral" where doctors from higher level centers come to perform operations at lower-level centers). But, they were not able to operate on him (presumably due to his comorbidities and providing safe anesthesia), and so Joseph had to wait to get referred to the zonal hospital. Joseph was not alone in needing referral health care. To provide more context, Box 1 provides a list of patients and conditions for patients for patients in the camp requiring referral during the research period (from interviews and clinical record review).

Box 1. Example of delayed referral case (name changed for anonymity).

**STANDARD MEDICAL REFERRAL REFUGEE CAMPS IN KIGOMA REGION**

**Camp:** Nyarugusu      **Date:** 10/02/2021

**Name of patient:** Ruth Dayana    **Age:** 36 years    **Sex:** Female

**Religion:** Christian

**Medical Condition:** Breast discharge for 3 years

1. **History:** The patient presents with breast discharge from the nipples that started gradually accompanied with breast swelling and pain. The pain radiates to the chest and back. She has been treated several times without any improvement. No history of chronic illness.

2. **Physical Examination:** Alert, afebrile. Not pale, no jaundice, no cyanosis, no lymphadenopathy, no LLE. BP = 113/70 mmHg, PR = 75b/min.

3. **Investigations:** Hb = 12.3 g/dL

4. **Treatment so far provided:** Ibuprofen. Duphaston.

5. **Diagnosis:** Hyperprolactinemia secondary to pituitary tumor.

6. **Reason for referral:** For specialist re-assessment and further management.

7. **Treatment recommended:** As per specialist review.

8. **Benefits expected:** Improve quality of life.

9. **Prognosis:** Good

**Referral Doctor's Name:** ___________________________

**Abbreviations:** Hb = hemoglobin. BP = blood pressure, PR = pulse rate (heart rate)

For patients like Joseph and others, sometimes the origin of their medical problems began either before or during their flight from their home countries.

## Problems began back home

Thomas, who was shot in Congo and whose anecdote introduced this article, is but one example of refugees whose problems began before crossing in Tanzania. Gloria, a middle-aged woman who had both a thyroid disorder and a mass on the back of her neck also said her problems began in Burundi before coming to Tanzania. Similarly, Christina, a middle-aged Congolese refugee, recounted her problem that has been ongoing for four years.

From Bukavu in eastern Congo, Christina was home one evening in 2017 when a loud knock was heard at the door. Rather than it being her husband she was expecting that day to return from a business trip, instead it was soldiers. They demanded money from Christina's mother, to which she replied she had none. The soldiers asked "How come you say you do not have money when your son is doing business?" Christina says the quarreling continued, and then a gun shot was heard. It was her mother that was shot. She died.

Next, two soldiers entered her room. "They grabbed me and wanted to have sex with me by force. I tried to refuse but they were too strong." The next thing Christina remembers was waking up in a hospital bed. Despite her husband going to report the incident, Christina noted that the soldiers returned to the house to kill her to "erase all the evidence." Fearing for her life, she fled to Tanzania.

Her medical problem had already started prior to entering Tanzania. "I was already having problems that were making my life hard. . .The problem I have is that urine passes uncontrollably. . ..even when I cough urine passes, or when I laugh urine passes." Christina has a fistula—an anomalous connection between her bladder and her vagina. Not an uncommon problem after early childbirth or rape, fistulas can be life-limiting problems. For Christina, it has caused "challenges" in her marriage. "It is like I am divorced," she says. "My husband has tolerated much, but now he is saying for how long is he going to tolerate this condition?" Her husband has found another wife. For patients like Christina, there are no surgical options in the refugee camp, so they must rely on being referred to other hospitals in Tanzania.

## "By waiting for referrals, patients continue to deteriorate"

Another common theme that arose in interviews and was confirmed with participant observation and clinical record review was that waiting was a common phenomenon for refugees. This took four primary forms: 1) waiting for an original referral to be written for a problem either to obtain a diagnosis or treatment; 2) waiting for transportation to be arranged to attend to the referral hospital; 3) waiting a long time at a referral hospital without a clear plan or improvement; 4) waiting to return to a referral hospital for a follow-up visit. Delays in the camp (waiting to be given a referral and waiting for transportation to the referral hospital) seemed to be the most significant—time wise. Joseph, the 23 year-old with heart failure and HIV described it as a mismatch between what is said and what is done:

> "They can tell you that you will be referred but to get that referral is where the problem is. You can stay 3 months or even 4 months without being referred. That is where you find someone ends up with complications."

These delays were not benign and had significant consequences for patients. In the words of Joseph again:

"The challenge I see there at hospital is delaying to refer patients. They know they don't have good services sometimes at all. You find someone needs an urgent referral but you will find them still not doing what they should do. A patient gets worse and the condition deteriorates. And sometimes it may be that that disease would have treated and cured but it ends in complications which are untreatable or irreversible and some people end up dying because of not being referred on time."

Joseph's comments were representative of many individuals we interviewed with regarding significant waiting periods even before initial referral. Part of this was in relation to the fact that the medical referral committee (MRC) typically meets only once per month and sometimes must sift through easily over 150 cases.

Clinical records showed that patients wait weeks to months to even receive approval at the meeting. Since many patients are batch presented at a monthly referral committee meeting, one's case may get discussed only once per month if not deemed urgent enough. Thomas' case is an example of this. Dates of consultation with a physician may vary and could be several weeks before the presentation at the referral committee meeting. The additional time to coordinate transportation (assuming it does not fall through as in the case of Thomas) and time for care to be delivered at the outside hospital can also take several weeks, leading to several months of delays for care. Another example of this is provided in Box 1, where a middle-age woman with a breast mass had her case presented at a meeting two months after she was seen to rule out cancer. This patient whom I refer to as Ruth is a 36 year old woman who had persistent drainage from her nipples. The concern was that she might have a brain cancer causing her nipple discharge. Despite being seen on February 10th, her case was not presented until two months later, and even after that, it would take days to weeks to organize the physical referral.

But in addition to waiting for an initial referral, other patients waited at an outside hospital for weeks on end without any progress in their care. For example, Christina, the 35-year-old with a vesicovaginal fistula, waited at the Maweni Regional Referral Hospital for weeks only to finally be transferred to the higher level of care directly from the regional hospital.

"I stayed at Maweni for about three weeks without any medication . . .they told me that let us not waste your time and to tell you the truth, for your problem there is no doctor here who can manage it. So, we have to give you a referral to Mwanza (BMC)."

Gloria, the middle-aged woman with a thyroid mass, also waited at the same referral hospital (Maweni) for several weeks. During the first three weeks, she was "having different tests" up until the 4th week when "they put [her] on the list to be operated on" only for her to arrive to the operating room and be told it was cancelled. It took several more weeks for her to then be transferred to the zonal hospital in Mwanza (Bugando Medical Centre). Taken together, Joseph, Christina, and Gloria are just a few of the examples of patients who were, on some level, waiting for referrals.

Finally, some patients despite having been referred to an outside hospital and being given an official follow-up appointment miss their appointment and/or wait to return to the outside hospital. For example, Marie, a 33-year-old woman with ongoing shortness of breath, leg swelling, and pain who had been treated at Maweni in Kigoma was—at the time of our interview—waiting for her return referral despite significant time passing by. Similarly, while Joseph was waiting for his return referral, "the date which was written for that follow up has already passed." From clinical record review or attending an MRC meeting, it was clear that other patients may have missed their follow-up appointment by the time the MRC meeting

was convened or during the meeting it was deemed not necessary for them to attend the follow-up appointment by the medical referral committee.

## "We Tanzanians are the first patients running to Nyarugusu camp"

In addition to the complexity of medical problems and different forms of waiting, another major theme that arose was a difference in health care seeking behavior and access between and among refugees and Tanzanians.

Importantly, findings from interviews, focus groups, and participant observation confirmed that services are offered free of charge. First, this included care given in the camp for refuges and costs associated with being given a referral. Second, this also included Tanzanian patients receiving health services in the camp, but for Tanzanians this did not include the cost of referral if one granted. While the focus of this article is on refugee reflections, one Tanzanian patient interviewed confirmed "Tanzanians are the first ones running to the camp" because "people are cared for and no payment [is required]." In other words, Tanzanian patients often went to the refugee camp instead of services outside of the camp because the services inside the camp were free of charge. Moreover, the resources available at the refugee camp dispensary may mirror or exceed those available at a district level hospital, despite the camp health center technically being a lower-level center in the Tanzanian hierarchy. This was in large part due to the resources and role of multiple NGOs. When it comes to referral at higher level health centers, though, Tanzanians must self-pay (or have insurance). On the other hand, for refugees, the cost of health care is free including referral health care. At the same time, though, refugees often reflected that some services offered in the camp were of poor quality or were unavailable (e.g. diagnostic tests). Therefore, several refugees indicated if they had the financial means to seek services outside of the camp, they would do so.

## "It's my life not theirs!"

As Thomas recounted his story of his chronic leg wound to us, we asked him if he had ever thought of leaving the camp without a permit to seek health care. He replied, "I had those thoughts several times, but the problem is money. I don't have any. I once thought of going to Burundi but at end of the day the problem remains the same: I don't have money." Upon probing further, we asked if money was not a problem, if he would leave the camp, to which he confirmed that of others we interviewed by stating: "Yes, if money were not a problem to me, I could just go without even waiting for the doctors in the camp because, life and health are mine, not theirs." He went on to re-tell a story of his neighbor who did the same:

> "There are a lot of other people also going out of the camp without having permission from MHA (Ministry of Home Affairs), it just depends on someone's economic means and what they are going to do. If someone gets sick, many of us just go out even without seeking a permit . . . because health is ours and not theirs.. . . For example, one of my neighbors was suffering from a hernia and when he went to the hospital, he was told there was no thread to sew. . . He fortunately has relatives in the U.S. So, he called them and told them that he was seriously ill and he is about to die. Then one of his relatives told him that you can't die there in the camp. He sent him money and he went to one hospital called Shunga, and he was treated well and now he is very fine. At first, he was not even able to eat well, but for now he eats anything and is completely fine. He is among those I know who went outside the camp to seek for medical treatment without a doctor's permit or MHA permit."

In this instance, a refugee who had the financial means was able to seek care for a basic surgical operation at a nearby hospital (Shunga) but did so without following the official referral process through the hospital or government. There was a distinction made between a referral from a physician and a permit from the Ministry of Home Affairs.

### Differences between refugees and Tanzanians

For refugees, a referral from a doctor and a permit from MHA are intertwined as legally they are not allowed to leave the camp without a permit from MHA, and a formal referral requires a referral from the physician. For Tanzanians, they may be treated in the refugee camp hospital and even provided a referral, but they do not require a permit from MHA to then travel to the higher-level health care center—even though a refugee and a Tanzanian may have the same exact diagnosis. Thus, there were far less delays for Tanzanian patients being treated in the camp, as they have the freedom of movement politically to leave the camp. This is in part a consequence or outcome of the refugee policies in Tanzania, and in doing so Tanzanians can get treatment or referrals in the camp and subsequently leave on their own accord to seek health care at a higher-level facility assuming they have the financial means to do so. At the same time, though, they are not subject to free services at the referral hospital that refugees would be. Of note, the restrictions and forced encampment policies had obvious impacts on the ability for refugees to participate in income-generating activities. Finally, while some refugees endorsed that they would indeed leave the camp had they the money, many reflected that despite living in the camp for years they had never left the camp or at most only travelled to a neighboring village.

## Discussion

For refugee populations who often receive health care at primary level facilities, referral health care is a major process for receiving adequate care for complex, or in this case, sometimes even basic medical or surgical conditions. In the context of a Tanzanian refugee camp, an examination of the referral process from the perspective of the refugees illuminates the complex problem space of seeking and receiving referral healthcare in a particular geopolitical context.

### Complex problems and sick bodies

While infectious diseases remain common in this context (e.g. malaria, upper respiratory infections, diarrhea), many refugees had complex problems, such as heart failure, a vesicovaginal fistula, cancer, or other chronic, non-communicable diseases. The lack of timely referral only exacerbated many of these conditions, as many refugees were not able to obtain care in a timely manner because they were not allowed to leave the camp. Complex problems were made even more complex because of delays in care.

In her seminal work *Casualties of Care*, Miriam Ticktin traces how exceptions are made on immigration policies for those with medical conditions [36]. She suggests that "unusual pathologies turn political" and that a compassion politics allows "sick bodies" to cross borders while "impoverished" ones cannot. In that space, the sick body becomes an apolitical subject, a biological citizen, morally worthy of treatment whereas poverty alone is not enough [29, 36]. In the context of the camp, even those with "sick bodies" though were not allowed to receive care in a timely manner and as Joseph aptly pointed out, this often led to a worsening of one's condition. Rather than one's being sick promoting an apolitical notion to treat suffering, for refugees in Tanzania there was an inherent political nature to their existence—the very origin and definition of refugees being political. While some sick bodies do eventually travel "across borders" to receive referral health care, many were subject to significant delays because of

underlying policies on freedom of movement. There was no such thing as an apolitical subject on medical grounds. Despite the refugee camp being governed by a transnational conglomeration of NGOs and the United Nations—in other words, a "regime of care like humanitarianism"—the Tanzanian state was still actively involved in approving and supervising the granting of permits for any refugee to leave the camp [37, 38].

## Three more delays of referral health care

In 1994, Thaddeus and Maine proposed the now well-known "Three Delays Framework" and its relation to maternal mortality [39]. In this model, three major types of delays contributed to maternal mortality including delays to seek care, delays in travelling to a facility, and delays in receiving adequate care. This model has had enormous impact on understanding health care seeking behavior in particular therapeutic ecologies.

In the context of referral health care for refugees in Tanzania, I suggest that there are at least three additional delays in referral healthcare including 1) delay in obtaining a referral from a physician even after reaching a health facility, 2) delay in getting to a referral hospital, and 3) delay in receiving care at the referral hospital. For some, the decision and process to pursue follow-up care (e.g. post-operative visits after a surgery) seemingly restart the cycle and delays of getting adequate care (Fig 1). For patients like Thomas, these delays were not just about health and the emotional toll waiting took on his life, but about life itself.

Understanding the individual factors that contribute to delays at each of these levels may have important public health implications. For example, patients may present to the hospital system but may not be referred initially, and it may take weeks for a patient to be written for a referral even after being admitted to a hospital (delay #1). This may in part be due to budgetary constraints known by the referring staff. Similarly, the same budgetary constraints may limit the number of referrals (and type) that occur in a given month or quarter. Then, there may be additional delays in organizing transportation and ensuring patients are appropriately informed about a referral (delay #2). Lack of equipment and resources at the higher-level hospital may contribute to delays in care (delay #3), and a lengthy bureaucratic process and

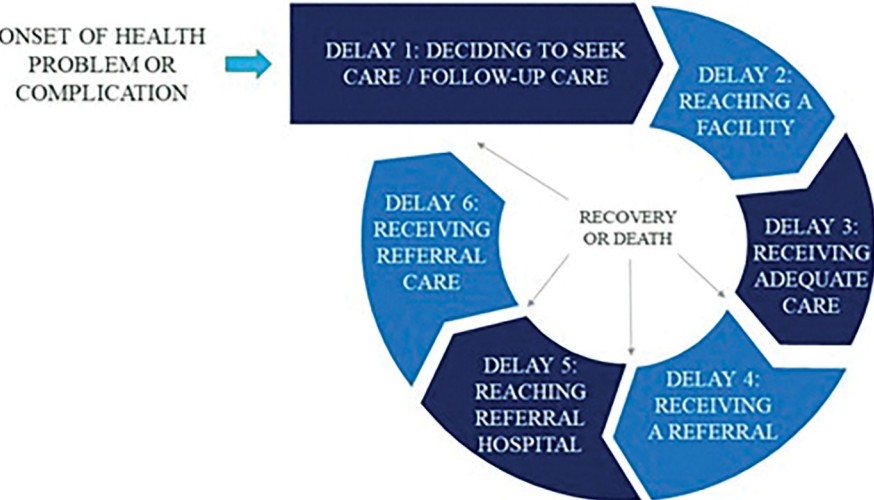

**Fig 1. Three more delays of referral health care.**

restrictions on movement may contribute to failure for patients to return for follow-up appointments (delay #4). Finally, difficulty in pursuing income-generating activities, particularly for those patients who would have sought health care outside of the formal referral process, lead to overall delays in referral health care, even if someone was willing to leave the camp without formal approval.

## On therapeutic itineraries

What happens, though, when one's problem is not being adequately managed at the lower-level center, such as the refugee camp hospital, particularly if one is a refugee and legally cannot leave the camp? The therapeutic itineraries of refugees in this context seemed to be secondary to both a geopolitical environment and economic one.

Itineraries imply movement, and for many refugees in this camp, their therapeutic routes or itineraries (i.e. health problems) started before or during their flight to Tanzania. Yet, once arriving in Tanzania, these therapeutic itineraries in a way become static secondary to the vast number of delays they face in accessing care. If refugees are in many ways defined by their mobility—leaving one nation-state to seek safety and security in another—then their ability to access referral health care may paradoxically be seen as immobile. Refugees in Tanzania are not legally allowed to leave the refugee camp without special permission from the government [7, 40, 41]. Referral for refugees is based on a particular set of conditions (individual, circumstantial, geopolitical) guided by both international and local standard operating procedures [22, 23]. What is more, recent scholarship in Tanzania has highlighted the concern of overburdening higher levels of care with problems that could be managed at potentially lower level centers [14, 42]. In other words, if a patient has a problem that could be managed at a district hospital, but self-presents to the national hospital, this could be a strain on already limited resources.

Tanzania practice states individuals should first be seen at a dispensary level and even recent anecdotal reports suggest some patients are being fined for self-presenting to higher level of cares without appropriate referral slips. Other anthropologists have described the ways in which patients' care seeking is at least in part due to a particular set of constraints or landscape for therapies available to them. For example, Stacey Langwick has traced how some patients combine both traditional medicine and biomedicine to improve their health [43]. Similarly, Dominik Matthes has traced how patients with HIV/AIDS have navigated opting for traditional medicines in place of antiretrovirals while still routinely attending clinic to test their CD4 counts [44]. This was in part due to work and economic considerations, as well as social ones. Finally, Vinh-Kim Nguyen in his seminal work *The Republic of Therapy* has traced how some HIV/AIDS patients enrolled in a clinical trial to gain access to life-saving medication in what he has called a therapeutic citizenship [28]. In applying these concepts of healthcare seeking behavior and therapeutic citizenship to the refugee camp, what emerges is a unique "therapeutic itinerary," where patients pursue particular health trajectories amidst particular sociocultural, economic, and political circumstances. Yet, for refugees these therapeutic itineraries, in reality, were actually more static than mobile given the series of delays experienced by refugees. Rather than being on a therapeutic journey from diagnosis to treatment, many of these patients remained in what might be best understood as a therapeutic purgatory.

Despite both receiving primary care at the same camp level facility, and despite if both had the same diagnosis, two individuals (one refugee, one Tanzanian) could have the exact same pathology yet starkly different and divergent therapeutic trajectories. Scholars have written about the problems of categorizing refugees and other vulnerable populations as simply passive actors in a larger macro and micropolitical matrix of health decisions, political economies, and

lived experiences. To borrow the term from Liisa Malkki, in this article, I suggest that refugees in this context, too, "categorized back" from that which was placed onto them as simple passive actors in a long, bureaucratic process of referral [45]. In other words, some refugees did state they had never left the camp or even would not in the event they had money to do so. At the same time, however, others such as Thomas emphatically reiterated that their health is theirs alone and not the property or decision of the government or others in positions of power. Seeking medical care in and of itself in some ways became an act of resistance against the label of passivity and the real and perceived forms of biopower in this refugee camp. While a specific diagnosis did not enable one to receive certain benefits from the state as has been observed in other, classical arguments of biological and therapeutic citizenship, but instead it compelled, even forced, some refugees to seek care outside the formal system.

## Conclusion, policy implications, and future directions

In this article, I have argued that from the perspective of refugees who require referral health care, there are often significant delays in realizing such care, despite it being free of charge at the camp level and referral hospitals. These delays are in many ways predicated on geopolitical and economic reasons, rather than simply medical or surgical ones. In other words, for refugees to obtain adequate health care, the problem is as much a geopolitical or economic one as it is a medical one. For many refugees, the unique set of circumstances either led to long delays in care or an exclusion from timely care altogether, for which some responded by seeking care elsewhere outside of the formal system. On the other hand, Tanzanians may also benefit from the medical facilities in this camp, just as refugees have historically benefited from existing health systems to care for complex problems [20, 46]. Globally, refugees number over 25 million, many of which must rely on host country populations and health care infrastructure to seek health care.

While standard operating procedures and other regulations do take this into consideration, on a microlevel, more attention should be paid to better delineating rate limiting steps in the referral process or better leveraging mobile technology to track or organize referrals. An example of this might be the use of mHealth messaging systems to inform patients of the day, time and date of a planned referral, so patients such as Thomas in this article, do not miss their chance that may not come for another several weeks or months. Communication about referral approval, timing, and feedback to and from the receiving hospitals is crucial to an effective health care system. Current technology and electronic medical records could be further harnessed to improve such communication and patient care. On a macrolevel, more attention might be paid to better delineating or revisiting policies of restrictions on freedom of movement that have major implications for timely access to health care. In the age of the re-formed East African Community to which both Tanzania and Burundi are parties to, one might revisit a more open concept of freedom of movement that might at worst produce opportunities for refugees to seek health care more independently, and at best actually allow refugees in this context to realize their right to health care.

## Acknowledgments

We would like to acknowledge Tanzanian Red Cross Society and field staff for the opportunity to pursue this research. The project was supported by a grant from the Association of Academic Surgery Global Surgery Research Fellowship and the Ben Kean Fellowship from the American Society of Tropical Medicine and Hygiene. The funding sources did not provide direct oversight or play a role in the data analysis. All views expressed in this article are the views of the author alone.

## Author Contributions

**Conceptualization:** Zachary Obinna Enumah.

**Data curation:** Zachary Obinna Enumah.

**Formal analysis:** Zachary Obinna Enumah.

**Funding acquisition:** Zachary Obinna Enumah.

**Investigation:** Zachary Obinna Enumah.

**Methodology:** Zachary Obinna Enumah.

**Project administration:** Zachary Obinna Enumah.

**Resources:** Zachary Obinna Enumah.

**Software:** Zachary Obinna Enumah.

**Supervision:** Zachary Obinna Enumah.

**Validation:** Zachary Obinna Enumah.

**Visualization:** Zachary Obinna Enumah.

**Writing – original draft:** Zachary Obinna Enumah.

**Writing – review & editing:** Zachary Obinna Enumah.

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
