## [Decision Letter · Decision Letter 0]

13 Dec 2022

PGPH-D-22-01641

“It’s my life, not theirs!” Therapeutic itineraries and refugee reflections on referral health care in western Tanzania

Dear Dr. Enumah,

Thank you for submitting your manuscript to PLOS Global Public Health. After careful consideration, we feel that it has merit but does not fully meet PLOS Global Public Health’s publication criteria as it currently stands. Therefore, we invite you to submit a revised version of the manuscript that addresses the points raised during the review process.

The reviewers suggest that greater details should be provided around the methodology and the methods that were used in the study. I therefore invite you to pay particular attention to this area while addressing all the other comments that the reviewers provided.

We look forward to receiving your revised manuscript.

Kind regards,

Ferdinand Mukumbang, PhD

Academic Editor

Journal Requirements:

1. Please provide additional details regarding participant consent. In the ethics statement in the Methods and online submission information, please ensure that you have specified what type you obtained (for instance, written or verbal, and if verbal, how it was documented and witnessed). If your study included minors, state whether you obtained consent from parents or guardians. If the need for consent was waived by the ethics committee, please include this information.

a. Please clarify all sources of funding (financial or material support) for your study. List the grants (with grant number) or organizations (with url) that supported your study, including funding received from your institution. 

b. State the initials, alongside each funding source, of each author to receive each grant.

c. State what role the funders took in the study. If the funders had no role in your study, please state: “The funders had no role in study design, data collection and analysis, decision to publish, or preparation of the manuscript.”

d. If any authors received a salary from any of your funders, please state which authors and which funders.

3. Please provide separate figure files in .tif or .eps format.

4. We have noticed that you have uploaded Supporting Information files, but you have not included a list of legends. Please add a full list of legends for your Supporting Information files after the references list. 

5. In the online submission form, you indicated that "Data used and analyzed in the current study are not publicly available due to privacy and personally identifiable health information that could compromise the participants who were interviewed in the study. Data could potentially be de-identified upon reasonable request but data was obtained in Kiswahili language.". All PLOS journals now require all data underlying the findings described in their manuscript to be freely available to other researchers, either 1. In a public repository, 2. Within the manuscript itself, or 3. Uploaded as supplementary information.

Additional Editor Comments (if provided):

Reviewers' comments:

Reviewer's Responses to Questions

**Comments to the Author**

1. Does this manuscript meet PLOS Global Public Health’s publication criteria? Is the manuscript technically sound, and do the data support the conclusions? The manuscript must describe methodologically and ethically rigorous research with conclusions that are appropriately drawn based on the data presented.

Reviewer #1: No

Reviewer #2: Yes

2. Has the statistical analysis been performed appropriately and rigorously?

Reviewer #1: No

Reviewer #2: N/A

3. Have the authors made all data underlying the findings in their manuscript fully available (please refer to the Data Availability Statement at the start of the manuscript PDF file)?

Reviewer #1: No

Reviewer #2: No

4. Is the manuscript presented in an intelligible fashion and written in standard English?

Reviewer #1: Yes

Reviewer #2: Yes

5. Review Comments to the Author

Reviewer #1: An important topic and very relevant study, however, due to the wide differences in country specific policies on refugee freedom of movement, and the impact of these policies on the answers to the research questions, more specific methodological details are required before the manuscript will add to the body of knowledge in this field.

Some specific questions are: What were the selection criteria for the participants and what were the demographic specifics of the participants? Why were only two Tanzanians selected if the author wanted to make comparisons with the standard of care for citizens. In fact, the comparison detracts from the study as it is not explored in enough detail. I would remove the comparison and focus on the conditions for refugees and compare in more detail to the policies and lack of adherence to the policies.

The Methods section requires greater specific details in order to prevent the self-confessed bias of the author. The data collection methodology and data analysis need more rigorous description (e.g. research assistant’s skills and training for data collection, analysis methods and findings from the memos and summaries, reliability and validity methods used for both data collection and analysis).

Since the findings of this study can only be applied in specific refugee contexts, it becomes very important to strengthen the conclusions derived from the study. Therefore, a very clear link between each research question and the answers found in the study and then most importantly, the policy and practice implications of the findings would be necessary if this study is to address the gap in the field/topic. A Policy brief to the Tanzanian government should be the obvious outcome of this study and therefore evidence of more rigorous qualitative scientific methodology needs to be evident in this manuscript.

Reviewer #2: This is an original idea on an area of which we have little knowledge or understanding. How refugees experience referral and transfer to care is of meaningful concern for the efforts, convolutions and potential delays that can impact on health outcomes. The manuscript is well written and the narratives are compelling. The opening section is particularly well-written and captivating.

A few small comments:

1 - BOX-1 contains what appears to be detailed medical notes of a particular patient. The author indicates that her name has been changed, however, the details appear to include address and location of her camp, as well as detailed health information. I am not sure this is consistent with confidentiality requirements, especially since the the study data is not made available to protect patient identities. If, however, the information is invented for the purposes of the study, I suggest cutting most of it, since addresses and patient numbers are not relevant to the story being told.

2 - A bit more detail about the methods and design of study would be welcome. Not much is needed for space, but it is not clear if this was a partly ethnographic study, or if it is a grounded theory approach to a qualitative study etc.

3 - Although the author indicates that care for non-refugee patients would be very different, it would be helpful to understand if the referral delays and problems are exclusively the experience of refugees or if community members attending the same hospitals have similar problems? Are the issues unique to refugee experience? Are local patients experiencing different issues, such as finding the funds to travel too a hospital in another district?

4 - I know that in other communities, e.g. Jordan, there is emphasis on parody so that refugees are never seen to receive more than local communities. It would be interesting to draw some high level comparison with other settings if possible.

Overall this is a welcome contribution to literature on refugee health and continuity of care for refugee populations. I strongly support publication of this paper and look forward to being able to share it with others.

6. PLOS authors have the option to publish the peer review history of their article (what does this mean?). If published, this will include your full peer review and any attached files.

**Do you want your identity to be public for this peer review?** For information about this choice, including consent withdrawal, please see our Privacy Policy.

Reviewer #1: No

Reviewer #2: **Yes: **Lisa Schwartz

---

## [Editor Report · Decision Letter 1]

21 Apr 2023

“It’s my life, not theirs!” Therapeutic itineraries and refugee reflections on referral health care in western Tanzania

PGPH-D-22-01641R1

Dear Zachary Obina Enumah,

We are pleased to inform you that your manuscript '“It’s my life, not theirs!” Therapeutic itineraries and refugee reflections on referral health care in western Tanzania' has been provisionally accepted for publication in PLOS Global Public Health.

Best regards,

Ferdinand Mukumbang, PhD

Academic Editor